# Study of Statistical Text Representation Methods for Performance Improvement of a Hierarchical Attention Network

Adam Wawrzyński * and Julian Szymański *

Department of Computer Systems Architecture, Gdansk University of Technology, 80-233 Gdansk, Poland
* Correspondence: wawrzynski.adam@protonmail.com (A.W.); julian.szymanski@eti.pg.edu.pl (J.S.)

**Abstract:** To effectively process textual data, many approaches have been proposed to create text representations. The transformation of a text into a form of numbers that can be computed using computers is crucial for further applications in downstream tasks such as document classification, document summarization, and so forth. In our work, we study the quality of text representations using statistical methods and compare them to approaches based on neural networks. We describe in detail nine different algorithms used for text representation and then we evaluate five diverse datasets: BBCSport, BBC, Ohsumed, 20Newsgroups, and Reuters. The selected statistical models include Bag of Words (BoW), Term Frequency-Inverse Document Frequency (TFIDF) weighting, Latent Semantic Analysis (LSA) and Latent Dirichlet Allocation (LDA). For the second group of deep neural networks, Partition-Smooth Inverse Frequency (P-SIF), Doc2Vec-Distributed Bag of Words Paragraph Vector (Doc2Vec-DBoW), Doc2Vec-Memory Model of Paragraph Vectors (Doc2Vec-DM), Hierarchical Attention Network (HAN) and Longformer were selected. The text representation methods were benchmarked in the document classification task and BoW and TFIDF models were used were used as a baseline. Based on the identified weaknesses of the HAN method, an improvement in the form of a Hierarchical Weighted Attention Network (HWAN) was proposed. The incorporation of statistical features into HAN latent representations improves or provides comparable results on four out of five datasets. The article presents how the length of the processed text affects the results of HAN and variants of HWAN models.

**Keywords:** natural language processing; text representation; document classification; deep learning

## 1. Introduction

Language is a natural way of exchanging information between people. It is a fast and convenient way of communicating, which explains the popularity of instant messengers used by millions of users every day. It is also a way to store and exchange business, government, medical and research data in form of text documents. Because of its universal usage, humankind is generating a vast amount of text data every day due to the usage of the Internet.

The complex nature of human language makes it hard for computers to process in its raw form. To effectively use an increasing number of documents, it is necessary to transform text to machine-processable format, that is, into the form of numbers. Currently, a prominent way of creating numerical representation of documents is to map text into a geometrical space in such a way that different documents are far away from each other and similar ones are close. All algorithms described in this survey built upon this assumption.

The motivation for the study was to find out to what extent the use of deep neural networks improves the quality of representations of texts and how the improvement of quality influences computation time. Some papers aim to study the influence of different text representations [1–3]. An early study on using neural networks for text representation was presented in [4]. A more recent and extensive survey was presented in [5], including a wide theoretical review of existing algorithms, but lacking a comparison of the effectiveness

of the algorithms for selected datasets. In other studies, evaluation has been performed for texts oriented in a particular domain [6,7].

The contribution of this paper is the proposed HWAN method, which extends the HAN model by including statistical features of the text in the latent representation, thus improving the text classification performance in four out of five datasets. The experiments conducted show that it is possible to improve the quality of document representation by incorporating statistical features learned on a large external corpus.

This paper is structured as follows: In Section 2.1, natural language processing methods are described. In Sections 2 and 3 consecutively, statistical models and deep neural networks are presented. State-of-the-art models are presented in Section 4. Section 5.1 describes the datasets used for the evaluation of the presented algorithms. The experiment setup is presented in Section 5 and comparison results in Section 6. Section 7 presents our approach for the improvement of a Hierarchical Attention Network that is state-of-the-art for the neural representation of a text. Section 8 discusses the achieved results. We finalise our paper with the conclusions section.

## 2. Statistical Models

We start our survey with a brief description of text preprocessing methods. We then present text representation methods based on statistical algorithms.

### 2.1. Natural Language Preprocessing

A document may be considered a sequence of characters not separated by punctuation marks or spaces, which is an obstacle to the further processing of the document. For this purpose, text segmentation methods are used, which involve partition of the text into words and punctuation marks or dividing documents into utterances. In general, for languages with specified utterance boundaries, segmentation can be performed by splitting texts by punctuation marks or apostrophes, but there are cases where this simple rule does not work, for example, with abbreviations or possessives. This process is essential for most text representation algorithms that operate on single words or tokens.

The noise can be reduced by limiting the size of the dictionary. This can be achieved by different methods (described further), but during preprocessing it can be done by simplifying different inflexion forms of the same word. Three methods are used for this purpose: morphological analysis, lemmatisation and stemming. Morphological analysis is a method of natural language analysis, which provides information on grammatical properties and brings words into basic forms. The second method is based on simplifying words by reducing their inflected forms, for instance, "better" → "good". This method is usually used when processing languages with rich flexion, for example, Polish. The last method of reducing vocabulary size is stemming, which brings words into their basic form, for example, "running", "runs", "ran" → "run".

Documents are usually gathered from different sources and in different data formats, such as XML and HTML. Thus, during processing, the format of texts should be unified. Usually, they contain special characters, for example, text decorations used on Internet forums or e-mail addresses, which should be removed to reduce the bias of machine learning models.

In natural language, some words are required by language syntax but they do not carry much information. They are often the most frequent words in a given language and are often removed from texts to reduce vocabulary size, accelerate computations and reduce bias. Examples of stop words in English are: "the", "a", "an". The drawback of this step is spoiling the syntax properties of text.

### 2.2. Bag of Words

Bag of words (BoW) [8] is the simplest form of statistical representation of text. In this method, a document is represented by a set of words with associated occurrence frequency. An important drawback of this representation is an absence of word order and a lack of

context. To reduce the size of the resulting vectors, generally the following preprocessing steps are applied: stop words removal and lemmatization or stemming. To mitigate the lack of context, one can use n-grams instead of word frequencies, for example, bigrams or trigrams.

Because of its simplicity and speed, it is often used as a preprocessing stage in more advanced statistical methods. However, it is limited in processing complex relations between documents and exploiting semantic information.

BoW can be extended by using weighting schemes that relate documents with words. One of the most popular extensions is Term Frequency–Inverse Document Frequency (TFIDF) [9,10], a statistical method for computing the weights of words based on the frequency of their occurrence among documents in a dataset. For all words in the corpus, the ratio of inverse frequency to a number of all documents containing this word is calculated. Terms with the highest value of TFIDF weight usually characterize the document well.

### 2.3. Latent Semantic Analysis

Latent Semantic Analysis (LSA) [11] is a statistical method used for high dimensional data. It performs singular value decomposition (Appendix A) of the original matrix, a word–document co-occurrence matrix in the case of documents.his results in a more dense, lower-dimensional representation, and is called latent semantic space.

Let us denote a collection of $N$ documents $D = d_1, d_2, \ldots, d_N$ with words $w$ from dictionary $V$ of size $M$, where $V = w_1, w_2, \ldots, w_M$. By ignoring the order of words, we can represent those data in a word–document co-occurrence matrix $\Sigma$ of size $N \times M$ with values $Z = z(d_i, w_j)_{i,j}$. Function $z$ denotes the frequency of word $w$ in document $d$ or its TFIDF weight.

To compute the similarity between two words, the dot product of the $i$-th column and the $j$-th row of matrix $\Sigma$ is calculated because it contains words and their weights in latent semantic space. To calculate the similarity between two documents, the dot product of the $i$-th column and the $j$-th row of matrix $\Sigma V$ is computed.

The original co-occurrence matrix is usually sparse but vectors computed using singular value decomposition are dense. It is possible to create valuable representations of two documents even when they do not have common words, because synonyms and ambiguous words are transformed similarly. Due to dimension reduction of the representation vectors, similar words are merged and infrequent words are filtered out. This feature works only in case when the averaged meaning of words is close to the original. In other cases, it may introduce noise to the model.

### 2.4. Latent Dirichlet Allocation

Latent Dirichlet Allocation (LDA) [12] is an algorithm representing documents as a mixture of topics based on the probability of a word belonging to those topics. The method can be described in a few steps. For every document, parameters $\alpha$ and $\beta$ are computed according to the known a priori number of topics, denoted as $T$, where $\alpha$ is the probability distribution assignment of documents to topics and $\beta$ is the probability distribution of assignment of words to topics. Afterwards, for each document $d$ parameter $\theta_i$ that is the probability distribution of topic assignment in document $i$ is drawn from the Dirichlet distribution parametrized by $\alpha$. Next, for each topic parameter $\phi_t$, the probability distribution of word assignment to topic $t$ is drawn. Afterwards, for each word $j$ in each document $i$, topic $z_{i,j}$ is drawn from multinomial distribution parametrized by $\theta_i$.

In the beginning, it is necessary to prepare data for the LDA algorithm. A dictionary $V$ of size $M$ and a matrix of size $D \times \max(N)$ are created, where $D$ is a number of documents and $\max(N)$ is the maximum length of a document in terms of the number of words among all documents. The next step is to randomly assign topics to words and to calculate the frequency of assignments of each word to each topic. Then, a matrix of size $M \times T$ of the initial probability distribution of words over topics is computed. By analogy, the matrix of size $D \times T$ of the probability of assigning documents to topics is calculated.

Instead of sampling from multinomial distribution, the Gibbs sampling algorithm [12] is often used (Appendix B).

After the Gibbs sampling algorithm finishes, the probability distribution of the assignment of the document to one of $T$ topics can be calculated by summing a number of words assigned to each topic.

## 3. Deep Neural Networks

In recent years, the usage of neural networks for text representation has shown some advantages over the statistical approaches. In this section, methods based on deep neural networks are presented.

### 3.1. Partition-Smooth Inverse Frequency

Intuition suggests that longer spans of text could be represented based on a representation of each word and its context. Using pre-trained models, such as Global Vectors (GloVe) [13] or word2vec [14], we can use word representations, which retain much of the syntactic and semantic relations in their projection space. The document consists of multiple utterances that contain multiple words. This hierarchical structure is reflected in many algorithms that process long texts, for example, the Hierarchical Attention Network.

Averaging word vectors is a simple method for representing long text, but its drawback is a variety of words' semantic meaning. The resulting representation is the mean of all components, which for long texts provides similar points in multidimensional space. Averaging leads to a loss of syntax and semantic information encoded into words' representations. Partition-Smooth Inverse Frequency (P-SIF) is a method of weighted averaging according to a text topic, which mitigates these drawbacks.

In general, vectors averaging relies on replacing all words with their numerical representations. Afterwards, the concatenation of the dot product of every word with topic weights results in a matrix. Next, matrices are averaged, resulting in document representation.

Let us denote a corpus consisting of $D$ documents, a dictionary as $V$ and a weights matrix as $W$, which represents the probability of assignment of a word to one of the topics. Let us denote $p(w)$ as the probability of the occurrence of a word $w$ in the corpus and $K = 3$ as the number of topics. For every word $w$ from dictionary $V$, a vector of parameters of membership degree $\alpha_{w,1}, \ldots, \alpha_{w,K}$ to each topic $K$ is computed. Let us assume that document $d$ consists of word vectors $v_1, v_2, \ldots, v_n$ , where $n$ is the length of the document in number of words. Document representation, presented in Formula (1), is calculated as a concatenation of weighted sums of word vectors for every topic where operator $\oplus$ represents vector concatenation:

$$D = (\sum_{i=1}^{n} v_i \times w_{i1}) \oplus (\sum_{i=1}^{n} v_i \times w_{i2}) \ldots \oplus \ldots (\sum_{i=1}^{n} v_i \times w_{iK}) = \oplus_{k=1}^{K} (\sum_{i=1}^{n} v_i \times w_{ik}) \quad (1)$$

The resulting matrix has size $K \times M$, where $M$ is the word vectors dimension. According to simple word vectors averaging, which results in a $1 \times M$ matrix, the P-SIF algorithm results in a bigger dimension concerning document topics, which is important for long texts.

In the training procedure [15], first, word vectors from the dictionary are trained, followed by the creation of word topics vectors and, finally, the re-weighting process. Formula (2) presents how to compute a word vector for given sparsity parameter $k$ and upper limit $m$ for a list of normal vectors $\vec{A}_1, \vec{A}_2, ..., \vec{A}_m$, where at least $k$ parameters $\alpha_{w1}, ..., \alpha_{wm}$ are non zero and $\vec{\eta_w}$ denotes the noise vector.

$$\vec{v_w} = \sum_{j=1}^{m} \alpha_{wj} \vec{A}_j + \vec{\eta_w}. \quad (2)$$

In the first phase, sparse coding was used [16] but this step can be skipped by assigning pre-trained word representations, as was done in this study.

To train word vectors for each word $w$, $K$ different topic vectors $\vec{cv}_{wk}$ of size $M$ are created by computing the dot product of a word vector with parameters $\alpha_{w,k}$ for each topic, as presented in Formula (3).

$$\vec{cv}_{wk} = \vec{v}_w \times \alpha_{wk}. \tag{3}$$

Afterwards, the concatenation of all $K$ word topics vectors into matrix $\vec{tv}_w$, representing the topic–word relation of size $K \times M$, is calculated as presented in Formula (4).

$$\vec{tv}_{wk} = \oplus_{k=1}^{K} \vec{cv}_{wk}. \tag{4}$$

In the last phase, weights for words matrices $\vec{tv}_i$ are calculated according to Formula (5), using smooth inverse frequency formula $\left(\frac{a}{a+p(w)}\right)$, where parameter $a$ controls the shape of the weighting function.

$$\vec{v}_{d_n} = \sum_{w \in d_n} \frac{a}{a + p(w)} \vec{tv}_w. \tag{5}$$

Finally, common context from averaged document vectors is removed by deleting the first principal component, calculated by using PCA [17]. This operations is defined by the Formula (6), where $\vec{u}$ is the first singular vector:

$$\vec{v}_{d_n} = \vec{v}_{d_n} - \vec{u}\vec{u}^T\vec{v}_{d_n}. \tag{6}$$

### 3.2. Doc2Vec—Distributed Memory Model of Paragraph Vectors

Paragraph Vector is a method for the representation of longer, variable-length texts such as phrases, sentences or whole documents [18]. In this model, the task used for training is to predict the words in a paragraph. A concatenation of a paragraph vector, along with several vectors of words found in the context, is performed and then the most likely word matching the context is inferred, as presented in Figure 1. Paragraph vectors are unique, while word vectors are shared between paragraphs. The model template is a modified version of the Continuous Bag of Words model [19,20], which is computed according to Formula (7), where $U$ and $b$ are the parameters of the *softmax* function, $h$ is the concatenation or averaging function of the word vectors obtained from the matrix $W$ of words in the vocabulary, and the parameter $k$ is the size of the context window.

$$y = Uh(w_{t-k}, ..., w_{t+k}; W) + b. \tag{7}$$

Each paragraph is represented by a corresponding column in the matrix $D$, and each word vector is represented by a corresponding column in the matrix $W$. The paragraph vector and the context word vectors are then averaged or concatenated, and the next context word is predicted. An additional element that appears in the model is the paragraph token, which can be understood as the next word. It acts as a memory that stores missing words from the context of the paragraph topic. This is where the name Distributed Memory Model of Paragraph Vectors comes from. The size of the context is an application-specific parameter. The paragraph vector is shared among all contexts generated by the sliding context window on the paragraph. The word vector matrix $W$ is shared between paragraphs.

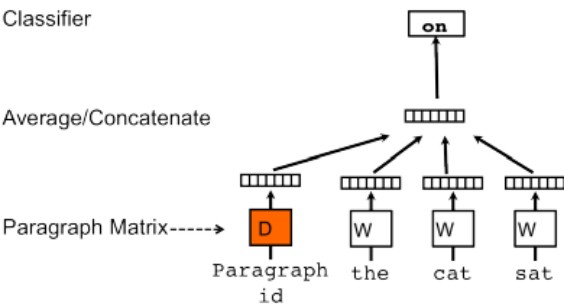

**Figure 1.** Distributed Memory Model of Paragraph Vectors [18].

The architecture of the model is based on a single-layer neural network. The input parameters are words from the context window in the form of a one-hot vector or as a dictionary mapped index and a paragraph index. The output parameter is the predicted word belonging to the context, analogous to Continuous Bag of Words.

The algorithm has two phases of operation: Phase one, which consists of training the word vectors $W$, and training the network weights and the paragraph vectors $D$, and Phase two, in which the model learns new paragraphs by adding their vectors to the matrix $D$ and training the network without updating the matrix $W$ or the network weights. Inference uses the resulting matrix of paragraph vectors $D$. An important feature of the model is the use of unlabelled training datasets. They inherit the vector semantics of the obtained representations from the Continuous Bag of Words model and also take into account the order and contextuality of the words.

### 3.3. Doc2Vec—Distributed Bag of Words Paragraph Vector

The Paragraph Vector Distributed Bag of Words method, analogous to Skip-gram, predicts the context based on some token. In the case of Skip-gram, this is the word central to the context and, in the case of the Distributed Bag of Words Paragraph Vector, it is the paragraph vector. In practice, we sample the context window then randomly select a word from the context and perform classification with the given paragraph, as presented in Figure 2.

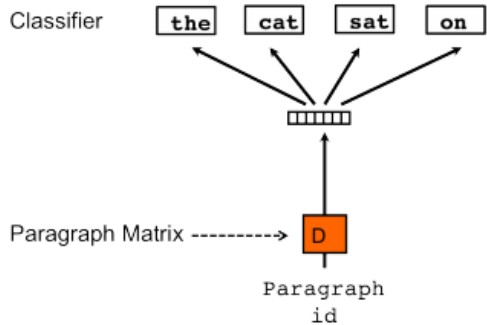

**Figure 2.** Distributed Bag of Words Paragraph Vector [18].

The architecture of the model is based on a single-layer neural network. The input parameter is the paragraph vector and the output parameters are words belonging to the context, analogous to Skip-gram.

Unlike the Distributed Memory Model of Paragraph model, it is much simpler and requires fewer data to be stored but its performance is slightly worse [18].

### 3.4. Hierarchical Attention Network

Hierarchical Attention Network (HAN) [21] is a deep neural network model used to represent documents employing the hierarchical structure of documents. Due to the varying amount of information that words or phrases carry depending on the context, two levels of attention mechanism were used—first at the word level and second at the sentence level (Figure 3). The attention mechanism additionally carries the ability to monitor which word or phrase contributed the most to a particular decision made by the model, which helps us to understand decisions made by the model. The implementation used in our experiments was based on the publicly available implementation repository [22].

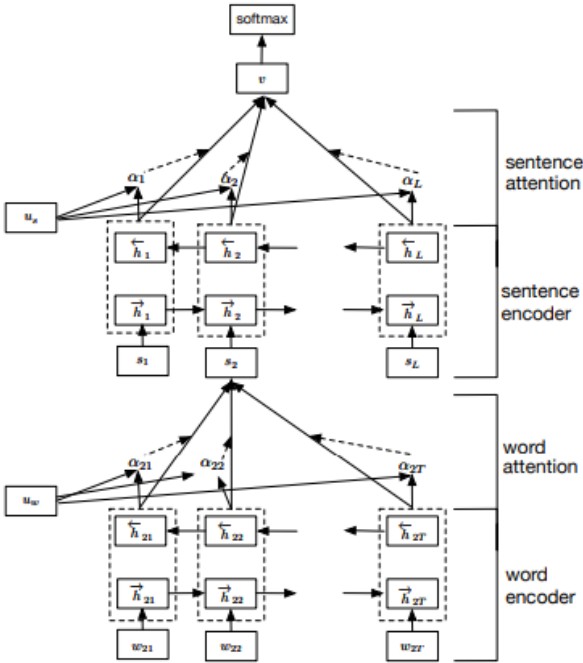

**Figure 3.** Hierarchical Attention Network [21].

The HAN model consists of a word encoder, a word attention layer, a sentence encoder and a sentence attention layer. Suppose a document contains $L$ sentences denoted by $s_i$, and each sentence contains $T_i$ words. The $w_{it}$ of the range $t$ in $[1, T]$ represents the words in the $i$th sentence. Given a sequence of word vectors $w_{it}$, $t$ in $[1, T]$, we perform their encoding by the matrix $W_e$, $x_{ij} = W_e w_{ij}$. Using bidirectional GRU networks [23], we obtain the encoded vectors by summing the output information from both directions, thus including the context information. Latent states of the model are calculated according to Formula (8). The bidirectional GRU layer contains a forward part $\vec{f}$, which, based on the sentence $s_i$, processes words from $w_{i1}$ to $w_{iT}$ and a backward part $\overleftarrow{f}$, which processes words from $w_{iT}$ to $w_{i1}$.

$$
\begin{aligned}
x_{it} &= W_e w_{it}, t \in [1, T] \\
\vec{h}_{it} &= \overrightarrow{GRU}_{x_{it}}, t \in [1, T] \\
\overleftarrow{h}_{it} &= \overleftarrow{GRU}_{x_{it}}, t \in [T, 1].
\end{aligned}
\tag{8}
$$

The encoded expressions are obtained as a concatenation of the hidden states of the network in both directions according to Formula (9).

$$
h_{it} = [\vec{h}_{it}, \overleftarrow{h}_{it}].
\tag{9}
$$

Because words affect the meaning of a sentence to varying degrees, an attention mechanism was introduced to extract keywords and aggregate them to form a sentence vector. Formula (10) presents HAN attention, where $u$ is the attention state for each word, $\alpha$ is its weight, and $s$ is the initial state of the encoded sentence obtained by summing the weighted states of the individual words.

$$
\begin{aligned}
u_{it} &= \tanh(W_w h_{it} + b_w) \\
\alpha_{it} &= softmax(u_{it}^T u_w) \\
s_i &= \sum_{t=1}^{T} \alpha_{it} h_{it}.
\end{aligned}
\tag{10}
$$

Initially, we feed the word representation $h_{it}$, obtained in the previous layer, into a 1-layer neural network to obtain its hidden state $u_{it}$. We then measure the relevance of the

words as the similarity of $u_{it}$ to the context of the words $u_w$ and obtain the weights $\alpha_{it}$ with the function *softmax*. We obtain the sentence representation $s_i$ as a weighted sum of the word representations $h_{it}$.

We obtain the encoded document vector in a similar way, using bidirectional GRU networks as presented in Formula (11).

$$\begin{aligned}
\vec{h}_i &= \overrightarrow{GRU_{s_i}}, i \in [1, L]\\
\overleftarrow{h}_i &= \overleftarrow{GRU_{s_i}}, i \in [L, 1].
\end{aligned} \tag{11}$$

We obtain an analogously encoded document as a concatenation of hidden network states in both directions according to Formula (12)

$$h_i = [\vec{h}_i, \overleftarrow{h}_i]. \tag{12}$$

Once again, we apply the attention mechanism, this time to encoded sentences (Formula (13)), where $u_s$ is the document context and $v$ is the resulting document vector. The only difference is the replacement of word representations with sentence representations.

$$\begin{aligned}
u_i &= \tanh(W_s h_i + b_s)\\
\alpha_i &= softmax(u_i^T u_s)\\
v &= \sum_{t=1}^{T} \alpha_i h_i.
\end{aligned} \tag{13}$$

### 3.5. Bidirectional Encoder Representations from Transformers

Transformer models are designed to pre-train a model on a large set of texts and then, using transfer learning, attach one output layer and tune it to solve a specific task. This way, we can easily obtain models that solve a wide range of problems without major architecture modifications and save resources. The unified architecture of the pre-trained model applied to different tasks is a distinguishing feature of the Bidirectional Encoder Representations from Transformers (BERT) model from other models whose architecture is tailored to the task being solved.

The BERT architecture consists of multiple Transformer layers created based on the original implementation presented by Ashish Vaswani [24].

The Transformer layer (Figure 4) is based on the encoder–decoder architecture but, unlike common recurrent networks, it uses feed-forward neural networks along with an attention mechanism. In the paragraphs below BERT, the base architecture is described.

The encoder consists of a stack of six identical layers. Each layer contains two sublayers: a multi-head self-attention mechanism and an element-wise fully-connected feedforward network. Each sublayer uses residual connections followed by layer normalization. Each sublayer has a projection to a 512-dimensional representation.

The decoder also consists of a stack of six identical layers. In addition to the two sublayers present in the encoder, a sublayer has been added that performs masked multi-head attention on the representations obtained from the encoder stack. In this sublayer, similarly to the encoder, residual connections have been applied along with layer normalization.

The attention model can be described as a vector of $n$ values that represent the mutual relevance of the words to each other. The multi-head attention (Figure 5) used in this model consists of eight parallel units. The first layer is a projection layer of $Q$, $K$, and $V$ to a 64-dimensional representation followed by a Scaled Dot-Product Attention function. The resulting vectors are then multiplied and a linear projection of their product is added to the final output value.

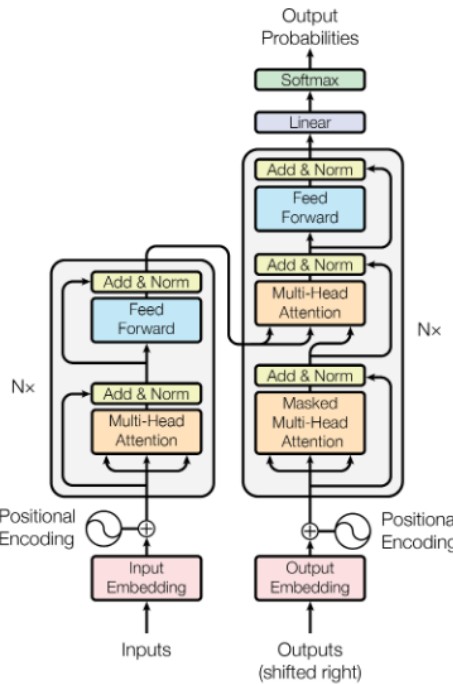

**Figure 4.** Transformer block architecture [24]. The left part is the encoder and the right part is the decoder.

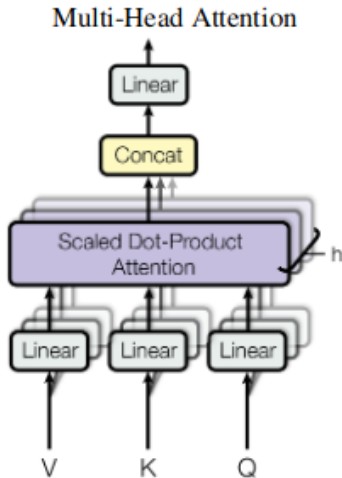

**Figure 5.** Multi-head Attention [24].

The linear projection of input word vectors is the product of the input vector with vectors of three weights resulting in vectors Query, Key and Value, denoted by *Q*, *K* and *V*, respectively.

The Scaled Dot-Product Attention function is computed for each input word vector (Figure 6). The attention value is calculated as follows: for each word, the matrix product between the vector $Q_i$ and the matrix of vectors *K* of each input word, including the Attention value for the currently processed word, is computed. The resulting value is then scaled by dividing by the squared root of the dimension of vector *K*. The value of the *softmax* function from the obtained vector of values for the attention value for the *i*-th word is then computed, and the product of the obtained values with the *V* vectors of all words is then computed. Finally, the sum of the obtained vectors represents the attention for the *i*-th word. In this way, attention is computed for all words. The dimension of the weights used to obtain the *Q* and *K* vectors must be equal, while the dimension of the *V* vector weights can be arbitrary. Scaled Dot-Product Attention is a function described by

the Formula (14), where $QK^T$ is the scalar product of vectors and $d_K$ is the dimension of vector $K$ (Figure 6).

$$Attention(Q, K, V) = softmax(\frac{QK^T}{\sqrt{d_K}})V. \tag{14}$$

We denote the number of Transformer layers as $L$, the size of the hidden layer as $H$, the size of self-attention heads as $A$, and the final architecture is taken as `L = 24, H = 1024, A = 16`.

### Scaled Dot-Product Attention

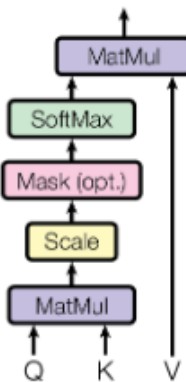

**Figure 6.** Scaled-Dot Product Attention [24].

*3.6. Longformer*

Longformer [25] is a deep neural network built from Transformer blocks [26] and optimized for the memory requirements. While the most common transformer model—BERT—has a quadruple relation input sequence length for memory requirements, Longformer has a linear relation between those parameters. This feature enables the usage of Transformer based models to handle long spans of text.

In order to optimize the memory consumption of the model, the attention function was modified, forming an attention pattern (Figure 7). Instead of fully computing the attention of each pair of tokens, a sliding window method was used, in which attention is computed for each pair of tokens that are in a window context where its length is less than the length of the entire input string. Additionally, a global attention mechanism was introduced. According to this mechanism, attention values for tokens with global attention are computed between those particular tokens and all tokens from the sliding window context. As a third possibility, random attention was introduced, in which tokens for which local attention is computed are randomized.

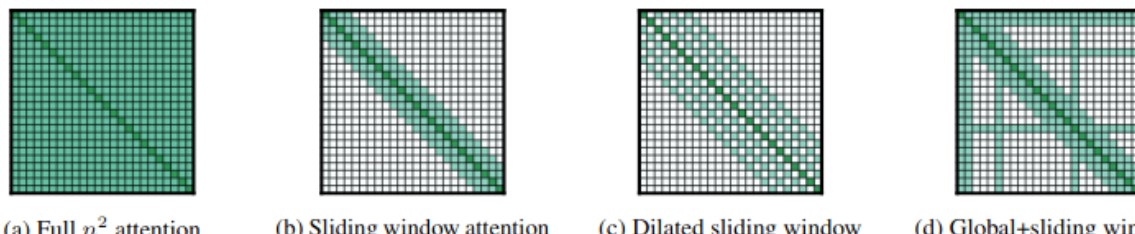

(a) Full $n^2$ attention     (b) Sliding window attention     (c) Dilated sliding window     (d) Global+sliding window

**Figure 7.** Different attention schemes [25]. Longformer attention scheme is presented by (**d**).

It is important to set up global attention correctly when using the model. The authors of the transformer's library [27] recommend using the global attention for the `<CLS>` token in text classification problems; for answering questions, the attention should be set to the

question tokens, and in language modelling, the attention should be set to the beginning of the utterance.

Using the optimized attention block, it is possible to process many times longer texts than using BERT. BERT-like models can process texts of 512 tokens in length, while Longformer can handle any length. In the case of the benchmark, a model processing 4096 tokens was used, but a model that handles up to 16,384 tokens has been also published [28].

## 4. State-of-the-Art Baseline

The current state-of-the-art models were obtained from the paperswithcode website [29], based on the best performance on the text classification subtask on diverse datasets.

Extensive use of recurrent networks to process sequential data was common before the Transformer era. An example of such a model is the bidirectional LSTM [30] network [31], which processes pre-trained word representations. A hierarchical variant of this model is the HDLTex network [32].

Most models use the Transformer architecture, which has been modified specifically for the task of document classification on shorter texts. Due to their growing popularity, the extension of these models, along with additional tricks, were used to process long texts more efficiently. A simple approach was to fine-tune the BERT model for document classification by including additional fully connected layers and a *softmax* cost function [33]. A similar approach was used in the paper "How to Fine-Tune BERT for Text Classification?" [34], where pre-training was additionally used on a training dataset, followed by fine tuning. Another architecture based on this architecture is the XLNet [35] autoregressive model, which uses two-stream self-attention for target-aware representation and, inspired by the TransformerXL [36], uses relative position encoding.

A different, yet very promising, graph approach is used in many current SOTA models, such as Massage Passing Attention Networks [37], which is the best model on the Reuters dataset. In this model, documents are represented as graphs built from a word co-occurrence matrix and a special node representing the document. In this model, a sentence representation is obtained by iteratively passing messages between nodes, and classification is performed on the glued representations of all sentences. A similar approach was presented in graph-based attention networks [38], which treat documents as nodes whose representations are vectors obtained by the bag-of-words method on which a self-attention operation is performed, together with neighbouring nodes. Similar assumptions were used in the Learnable Graph Convolution Network [39] model, in which an additional pre-learning step is the selection of a subgraph that allows the graph to be transformed into grid-like one dimensional data before training and the use of standard convolution layers. The best models were selected for comparison in the following section of the paper.

## 5. Experiment Design

To compare selected methods, one pipeline of training and evaluation for all models was designed. The first dataset is loaded into program memory. The second step is the preprocessing of texts. Then, a dataset is split into five parts following a K-fold cross-validation scheme. The model is trained on a training set of cleaned documents. The final evaluation is performed after the selected model generates a vector representation of the documents, which are passed into the classifier. The used classifier is a multi-class SVM algorithm [40], taking into account the prevalence of unbalanced training sets. After the procedure is finished, results are averaged and saved. As an evaluation metric, average overall accuracy was used, which is the percent of correct predictions provided by the model with standard deviation across all runs. The code for the experiment is publicly available [41].

The importance of text preprocessing was addressed in the paper [42], where the authors presented how much preprocessing affects the model output. In conclusion, they presented general findings: usually, tokenization achieves the same or better results than

more complex processing, such as coreference. The exception here was the Ohsumed collection, where there are many medical words, where the most efficient technique was to treat expressions or groups of words as a single token.

In our evaluation of all datasets following preprocessing, steps were applied: lowercasing (except Longformer, which was pre-trained on uncased texts), filtering email addresses and decorations.

The increase in the maximum length of the processed sequence is due to the increased VRAM requirements of the GPU cards on which the calculations were performed. The occurrence of outliers from the dataset that were many times longer than the mean would have unnecessarily increased the network size and increased the training time. Due to the limited hardware resources, it was necessary to limit the size of the neural networks in order to be able to perform the training. For models with a defined fixed length of the processed sequence of words and sentences resulting from the architecture of neural networks and their implementation details, it was decided to conduct the experiment in two variants. The first is to limit the length of the text to 30 sentences, each with a maximum of 100 words. The second is to limit the length based on the frequency distribution of sentences and words in the training datasets and, on this basis, to select the 99th percentile of the distribution.

The results of the training process for selected training datasets are presented below. The average accuracy and average training time of the models were measured.

The parameters with which the models were trained are as follows: For BoW and TFIDF, the dictionary size was not limited; `min_df = 1` and `max_df = 0.95` were selected as the number of minimum occurrences of a word and the maximum frequency ratio of a word in the set. Identical `min_df` and `max_df` parameters were also used in the LSA and LDA models. LSA was run with parameters `svd_features = 100`, which specify the number of dimensions after SVD reduction, the number of iterations `n_iter = 10`. Similarly, LDA, in which `n_components = 100` specifies the number of implicit topics to which words were assigned, and the number of iterations `epochs = 10`. The P-SIF model is parameterised by the number of clusters `num_clusters = 40` and the size of the resulting representation `embedding_size = 100`. Doc2Vec, DM and DBoW models were parameterized identically: the number of negative examples `negative = 10`, projection size `vector_size = 100`, context window `window = 5` and minimum word frequency `min_count = 1`. For the HAN model, the maximum number of sentences and words in sentences is configuration dependent. In addition to these parameters, the validation data ratio `validation_split = 0.1`, batch size equal to 16, and the number of epochs equal to 100 were used and an unlimited dictionary for statistical features was applied by parameter `max_features = None`. For the Longformer model, the number of iterations was set to `epochs = 10` and the batch size was equal to 4. The latter model was trained using full precision `fp16 = False` because of instability issues. As a pre-trained word vector, GloVe [14], with 100 dimension size, was selected.

Limiting the length of the processed text to the 99th percentile for each set was done by analyzing the frequency distribution of sentences and words in the sentences, from which the corresponding values were determined and used in the training process. Data regarding the state-of-the-art models were retrieved from the website [29].

## 5.1. Datasets

To evaluate selected text representation algorithms, five different datasets were selected [43,44] The selection of the datasets was based on a difference of the parameters: dataset size, document length, domain, the variance of document length and the variance in a number of samples in each class. The datasets used in the experiments are described briefly below.

### 5.1.1. BBCSport

BBCSport [43] consists of 737 documents about sports events published on the BBC website during the years 2004–2005. Texts are grouped into five classes: athletics, rugby, cricket, football and tennis. Data are not divided into train and test sets; therefore, a method of evaluation is cross-validation. Class frequency distribution is presented on Figure 8, whereas the distribution of word frequency is presented in Figure 9.

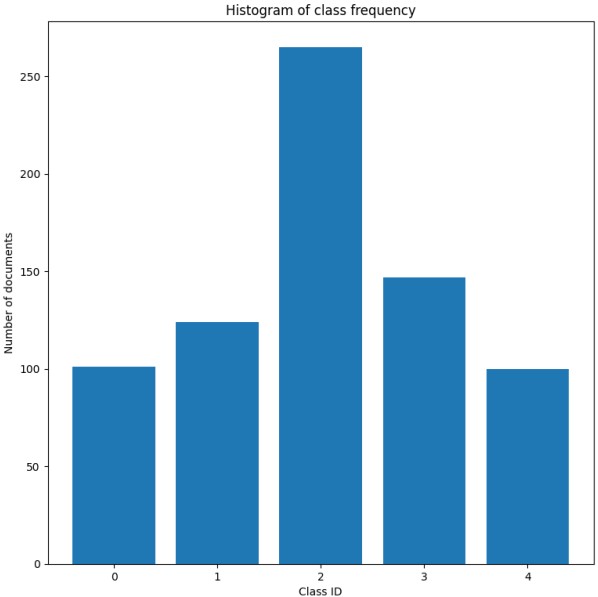

**Figure 8.** Class frequency distribution for BBCSport.

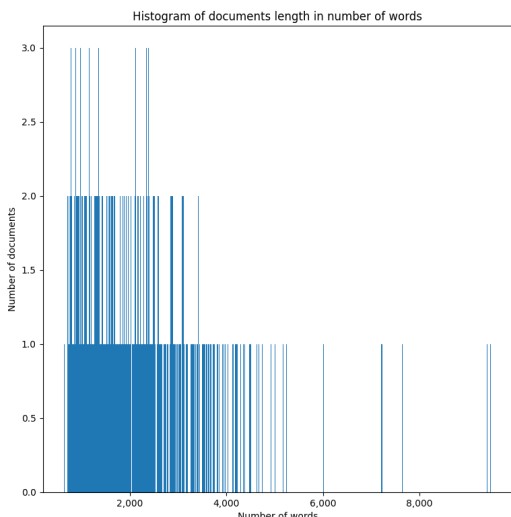

**Figure 9.** Distribution of words frequency in BBCSport.

The dataset is imbalanced in terms of the number of samples for each class and it has a shape of normal distribution. The most frequent class is class two, the rarest are classes zero and four.

### 5.1.2. BBC

BBC consists of 2225 documents published on the BBC website in the years 2004–2005. Texts are grouped into five categories: business, entertainment, politics, sport and tech.

Data are not divided into train and test sets, therefore a method for evaluation is cross-validation.

The dataset is balanced in terms of the number of samples for each class , as presented in Figure 10. According to Figure 11, the documents are in the range from 100 to 5000 words length.

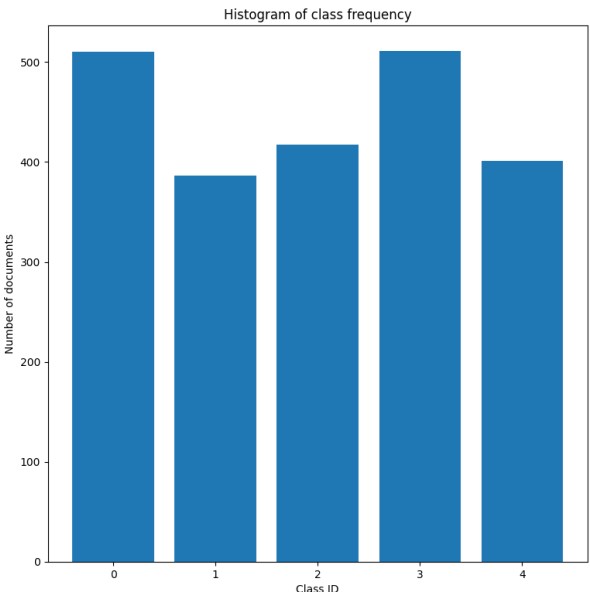

**Figure 10.** Class frequency distribution for BBC.

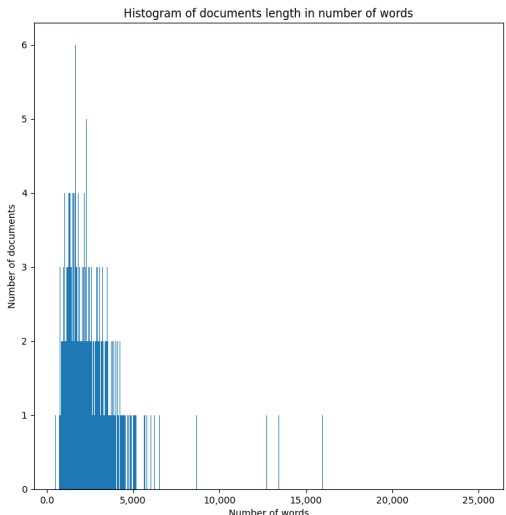

**Figure 11.** Distribution of words frequency in BBC.

### 5.1.3. Reuters

Reuters consists of 15,437 documents published on the Reuters website. Texts are grouped into 91 categories and the dataset is divided into train and test sets. Class frequency distribution is presented in Figure 8, whereas the distribution of the word frequency is presented in Figure 9.

The dataset is imbalanced in terms of the number of samples for each class. According to Figure 12, three classes dominate the dataset in terms of frequency. Documents are usually between 100 and 5000 words in length, as presented in Figure 13.

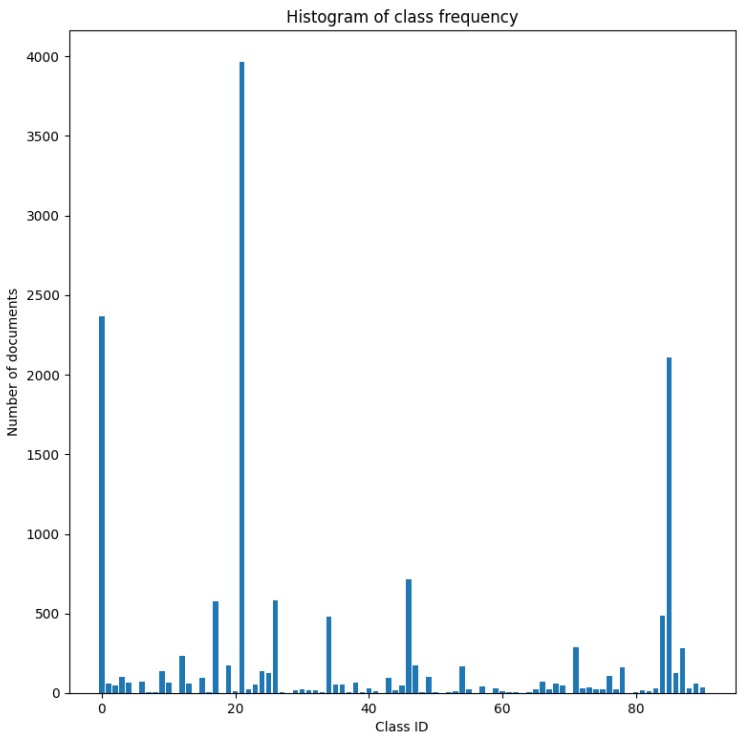

**Figure 12.** Class frequency distribution for Reuters.

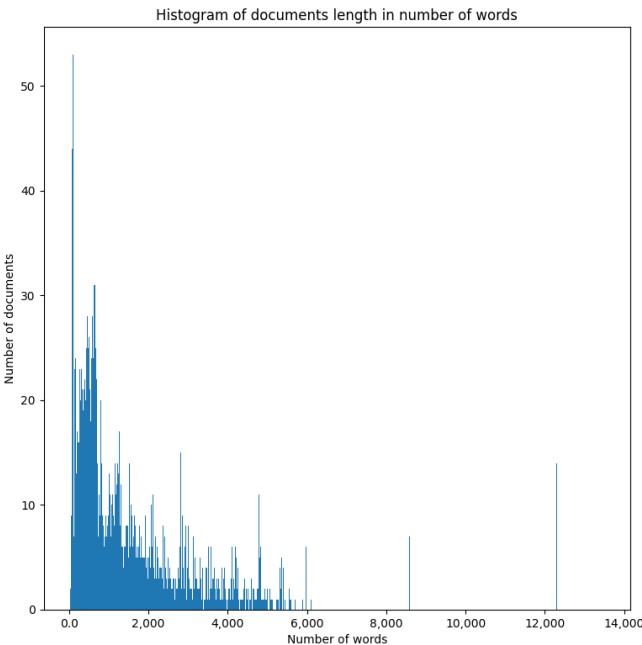

**Figure 13.** Distribution of words frequency in Reuters.

### 5.1.4. 20Newsgroups

20Newsgroups [45] consists of 20,417 documents from the Usenet website newsgroups collection. Texts are grouped into 20 categories and the dataset is not divided into train and test sets; therefore, the method of evaluation is cross-validation.

Documents are usually between a few and 10,000 words in length. Some samples are much longer than the average number of words, according to Figure 14. The dataset is balanced in terms of the number of samples for each class, as presented in Figure 15.

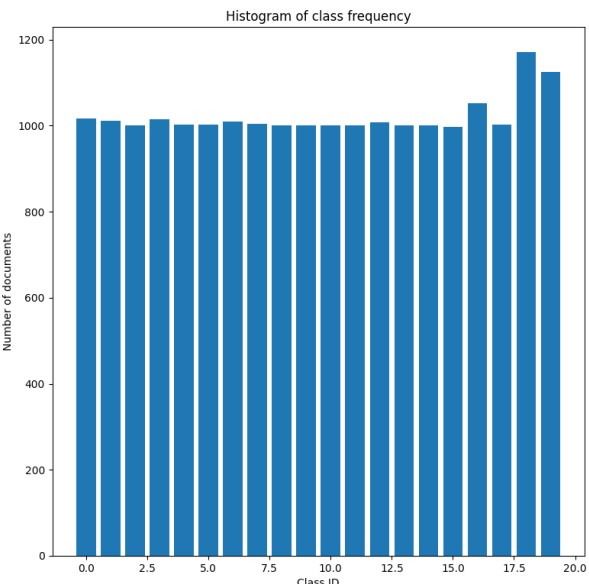

**Figure 14.** Class frequency distribution for 20Newsgroups.

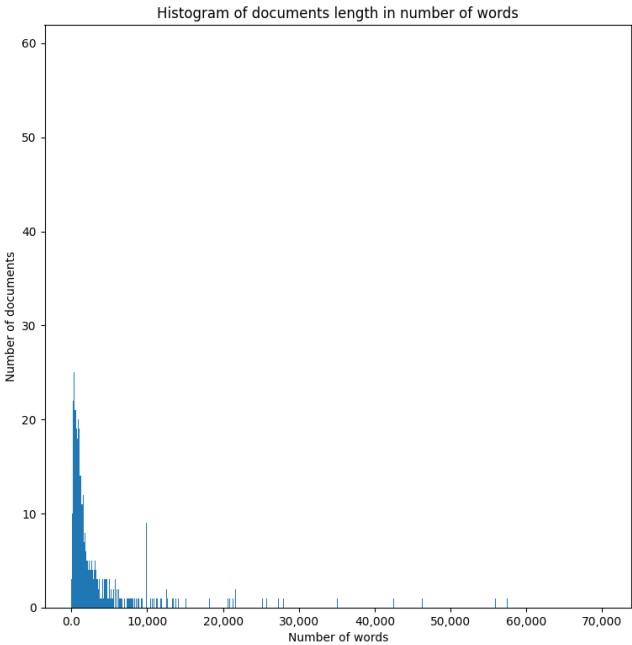

**Figure 15.** Distribution of words frequency in 20Newsgroups.

5.1.5. Ohsumed

Ohsumed [46] consists of medical abstracts from the MeSH category from the year 1991 and contains 20,000 documents divided into train and test sets. Texts are grouped into 23 categories of cardiovascular diseases.

The dataset is imbalanced in terms of sample number in each class. According to the chart in Figure 16, three classes dominate the dataset in terms of frequency, as presented in

Figure 16. The length of documents is usually between 200 and 3000 words, according to Figure 17.

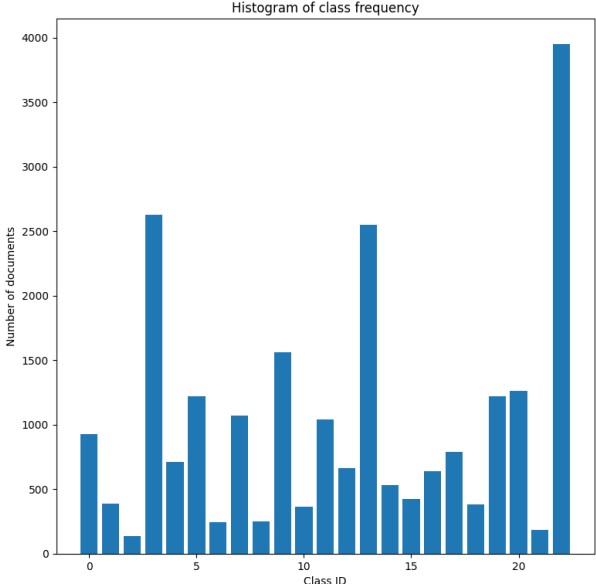

**Figure 16.** Class frequency distribution for Ohsumed.

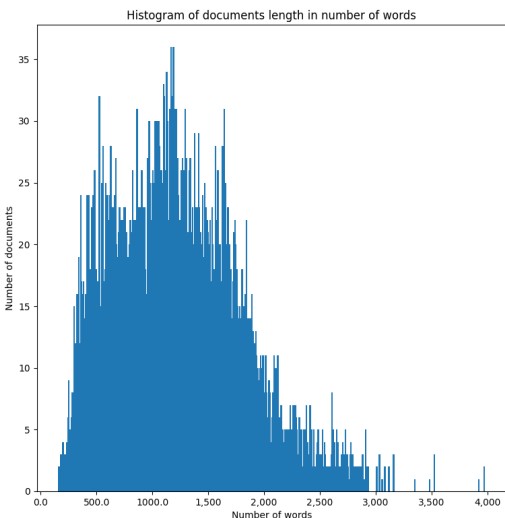

**Figure 17.** Distribution of word frequency in Ohsumed.

## 6. Results of Representations Comparison

In this section, detailed results of representations comparison in terms of averaged overall classification accuracy are shown in Table 1. Table 2 presents the training times required for training each model.

It can be seen from Table 1 that the statistical methods achieve results that are comparable to the selected neural methods. For the Reuters and Ohsumed collections, the statistical baseline achieved the best results, which are, however, significantly worse than the state-of-the-art. This may be related to the characteristics of the collections, such as the high unbalance of class examples. For five datasets—20Newsgroups, BBC and BBCSport—neural networks utilizing the attention mechanism outperformed the statistical baseline methods. For the BBCSport dataset model, HAN achieved 100% accuracy, outperforming the current state-of-the-art (based on the paperswithcode website).

Among the methods that performed better than the statistical baseline are the networks based on the attention mechanism—HAN and Longformer. Among these two methods, the HAN method achieved better accuracy and the model inference time is significantly shorter compared to the model based on the Transformer architecture (Table 2), and therefore this model was chosen for further analysis and modification.

**Table 1.** Averaged overall classification accuracy (in percent); cross-validation for Ohsumed, BBC and BBCSport, test–train split for Reuters and 20Newsgroups. The maximum length of input data for HAN was set at 30 utterances 300 words each. Numbers in bold denotes results better than BoW and TFIDF or the best overall result.

| Model | Reuters | Ohsumed | 20Newsgroups | BBC | BBCSport |
|---|---|---|---|---|---|
| BoW | **83.42** , $\sigma = 0.0$ | **60.25**, $\sigma = 0.0$ | 71.72, $\sigma = 0.0042$ | 97.03, $\sigma = 0.0077$ | 96.88, $\sigma = 0.0127$ |
| TFIDF | 75.58, $\sigma = 0.0$ | 56.44, $\sigma = 0.0$ | **85.55**, $\sigma = 0.0008$ | **97.84**, $\sigma = 0.0037$ | **98.51**, $\sigma = 0.0117$ |
| LSA | 64.12, $\sigma = 0.0017$ | 35.72, $\sigma = 0.0012$ | 71.76, $\sigma = 0.0053$ | 96.90, $\sigma = 0.0030$ | 97.83, $\sigma = 0.0117$ |
| LDA | 52.91, $\sigma = 0.0093$ | 20.07, $\sigma = 0.0036$ | 60.90, $\sigma = 0.0102$ | 93.08, $\sigma = 0.0056$ | 92.80, $\sigma = 0.0419$ |
| P-SIF | 62.61, $\sigma = 0.0016$ | 34.33, $\sigma = 0.0015$ | 63.41, $\sigma = 0.0057$ | 95.33, $\sigma = 0.0063$ | 94.70, $\sigma = 0.0267$ |
| Doc2Vec-DM | 79.81, $\sigma = 0.0013$ | 58.66, $\sigma = 0.0021$ | 75.25, $\sigma = 0.0075$ | 96.63, $\sigma = 0.0071$ | 97.29, $\sigma = 0.0096$ |
| Doc2Vec-DBoW | 82.65, $\sigma = 0.0006$ | 59.48, $\sigma = 0.0006$ | 82.52, $\sigma = 0.0040$ | 97.35, $\sigma = 0.0046$ | 97.96, $\sigma = 0.0096$ |
| HAN | 81.33, $\sigma = 0.0046$ | 54.39, $\sigma = 0.0029$ | **89.57**, $\sigma = 0.0066$ | **99.24**, $\sigma = 0.0046$ | **100**, $\sigma = 0.0$ |
| Longformer base | 79.90, $\sigma = 0.0042$ | 53.88, $\sigma = 0.0077$ | 81.33, $\sigma = 0.0054$ | **98.20**, $\sigma = 0.0028$ | **99.32**, $\sigma = 0.0061$ |
| SOTA | 97.44 [37] | 75.86 [47] | 88.6 [48] | | 99.59 [37] |

**Table 2.** Average total training and inference time in seconds.

| Model | Reuters | Ohsumed | 20Newsgroups | BBC | BBCSport |
|---|---|---|---|---|---|
| BoW | 27.16 | 20.42 | 68.50 | 5.79 | 1.20 |
| TFIDF | 0.95 | 0.95 | 2.31 | 0.54 | 0.11 |
| LSA | 25.51 | 45.82 | 38.70 | 8.48 | 1.88 |
| LDA | 363.29 | 179.01 | 947.30 | 193.08 | 69.32 |
| P-SIF | 363.14 | 356.04 | 777.95 | 164.63 | 71.26 |
| Doc2Vec-DM | 225.34 | 247.14 | 311.22 | 42.67 | 13.67 |
| Doc2Vec-DBoW | 183.42 | 190.18 | 241.81 | 34.31 | 11.94 |
| HAN | 3611.17 | 2816.97 | 5931.19 | 771.30 | 254.61 |
| Longformer base | 64,935.80 | 74,314.70 | 105,622.34 | 14,396.79 | 4919.38 |

## 7. Our Approach—Hierarchical Weighted Attention Network

By analysing the fine-tuned models using the Local Interpretable Model-agnostic Explanations (LIME) algorithm [49], insights were gained into how the best methods work.

Using the LIME algorithm, it was observed that a hierarchical attention network performs well with texts whose topic is not sewn into the statistical features of the text itself but is derived from the meaning of the text. However, there are cases where the HAN model assigns too much attention to certain words, as a result of which it incorrectly guesses the topic, while the statistical TFIDF approach performs correctly.

To overcome the weaknesses of the hierarchical attention network and to enrich them with statistical features, we propose the Hierarchical Weighted Attention Network (HWAN). The architecture of this neural network model is based on a hierarchical attention network enriched with the statistical values of a text's features.

Additionally, for each input token, the value of the statistical feature of a given word is passed as an input parameter of the model. For each word, its attention value is calculated, then, together with the statistical features (BoW, TF or TFIDF value for given words), a selected operation is performed on them (as shown in Figure 18 in the section "Word attention"). In the image, the statistical features are successively called $w_{21}$, $w_{22}$, up to $w_{2T}$, and the operation is denoted by the tensor product symbol. The result of this operation is a latent variable of a single sentence and it is passed to the document level bidirectional Long-Short Term Memory [30] layer. After that, there is the *softmax* layer, which calculates

the attention score for the whole document. Our study examined three different operations: concatenation (denoted in Tables 3 and 4 as *concat*), addition (*add*) and multiplication (*mul*) of word attention and statistical features. The models are named as follows: the first part of the name specifies the type of statistical feature and the second part specifies the type of information appending operation. Assuming that the statistical feature is BoW and the operation is *add*, we get the name BoW + add. The model parameters are the same as for the baseline HAN model.

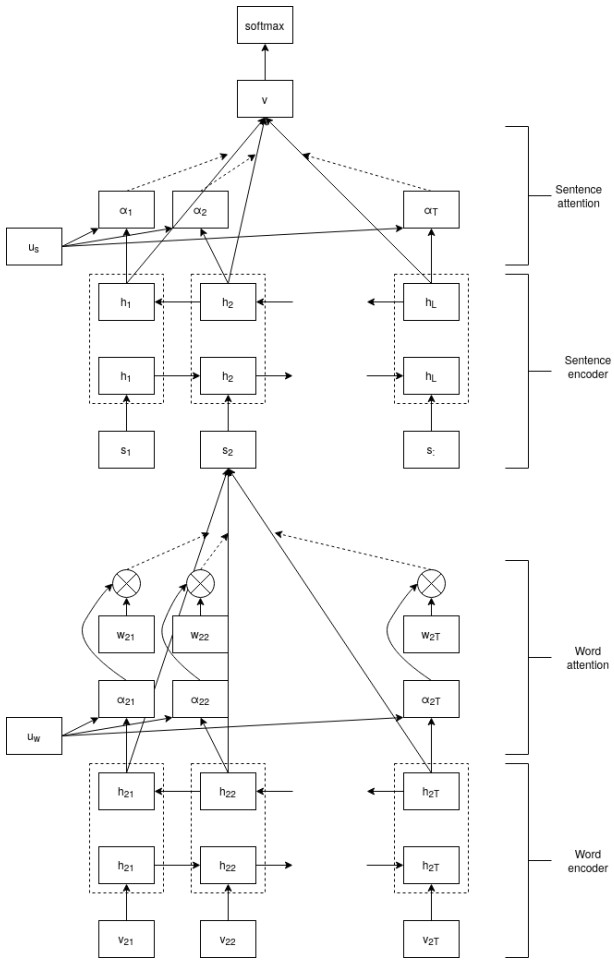

**Figure 18.** The architecture of our Hierarchical Weighted Attention Network.

**Table 3.** Averaged overall classification accuracy (in percent) for HAN and variants of HWAN; cross-validation for Ohsumed, BBC and BBCSport, test-train split for Reuters and 20Newsgroups. Restricting input length at 30 sentences 100 words each. Numbers in bold denotes results better than baseline HAN model.

| Model | Reuters | Ohsumed | 20Newsgroups | BBC | BBCSport |
|---|---|---|---|---|---|
| HAN | 81.33, $\sigma = 0.0046$ | 54.39, $\sigma = 0.0029$ | **89.57**, $\sigma = 0.0066$ | **99.24**, $\sigma = 0.0046$ | **100**, $\sigma = 0.0$ |
| TFIDF + add | **82.85**, $\sigma = 0.0022$ | **55.64**, $\sigma = 0.0027$ | 88.68, $\sigma = 0.0030$ | **99.69**, $\sigma = 0.0011$ | **100**, $\sigma = 0.0$ |
| TFIDF + mul | 72.76, $\sigma = 0.0051$ | 34.03, $\sigma = 0.0313$ | 70.51, $\sigma = 0.0168$ | 98.88, $\sigma = 0.0045$ | 69.51, $\sigma = 0.1957$ |
| TFIDF + concat | **82.44**, $\sigma = 0.0011$ | 53.33, $\sigma = 0.0044$ | 85.40, $\sigma = 0.0086$ | **99.64**, $\sigma = 0.0027$ | 99.59, $\sigma = 0.0054$ |
| TF + add | **82.81**, $\sigma = 0.0014$ | **55.39**, $\sigma = 0.0032$ | 88.68, $\sigma = 0.0038$ | **99.60**, $\sigma = 0.0026$ | 99.73, $\sigma = 0.0054$ |
| TF + mul | 73.50, $\sigma = 0.0057$ | 37.61, $\sigma = 0.0145$ | 72.41, $\sigma = 0.0088$ | 99.01, $\sigma = 0.0066$ | 87.29, $\sigma = 0.1613$ |
| TF + concat | **82.44**, $\sigma = 0.0017$ | 53.00, $\sigma = 0.0042$ | 85.86, $\sigma = 0.0051$ | **99.55**, $\sigma = 0.0038$ | **100**, $\sigma = 0.0$ |
| BOW + add | **82.88**, $\sigma = 0.0012$ | **55.10**, $\sigma = 0.0038$ | 88.01, $\sigma = 0.0131$ | 98.92, $\sigma = 0.0048$ | 98.51, $\sigma = 0.0079$ |
| BOW + mul | 81.97, $\sigma = 0.0011$ | 52.20, $\sigma = 0.0234$ | 71.51, $\sigma = 0.0079$ | **99.51**, $\sigma = 0.0030$ | **100**, $\sigma = 0.0$ |
| BOW + concat | **82.46**, $\sigma = 0.0017$ | 51.18, $\sigma = 0.0081$ | 75.08, $\sigma = 0.0101$ | 99.19, $\sigma = 0.0048$ | 99.32, $\sigma = 0.0043$ |

**Table 4.** Averaged overall classification accuracy (in percent) for HAN and variants of HWAN; cross-validation for Ohsumed, BBC and BBCSport, test-train split for Reuters and 20Newsgroups. Restricting input length at 99-th percentiles of the number of sentences and words. Numbers in bold denotes results better than baseline HAN model.

| Model | Reuters | Ohsumed | 20Newsgroups | BBC | BBCSport |
|---|---|---|---|---|---|
| HAN | 81.79, $\sigma = 0.0015$ | 51.19, $\sigma = 0.0061$ | 87.17, $\sigma = 0.0060$ | **99.55**, $\sigma = 0.0014$ | **100.0**, $\sigma = 0.0$ |
| TFIDF + add | **82.13**, $\sigma = 0.0016$ | 51.00, $\sigma = 0.0069$ | **87.43**, $\sigma = 0.0061$ | 99.01, $\sigma = 0.0046$ | 99.46, $\sigma = 0.0051$ |
| TFIDF + mul | 58.51, $\sigma = 0.0812$ | 25.15, $\sigma = 0.0404$ | 43.54, $\sigma = 0.3148$ | 79.60, $\sigma = 0.2893$ | 62.70, $\sigma = 0.3$ |
| TFIDF + concat | **82.06**, $\sigma = 0.0021$ | **51.66**, $\sigma = 0.0054$ | **88.14**, $\sigma = 0.0045$ | 98.79, $\sigma = 0.0046$ | 99.46, $\sigma = 0.0051$ |
| TF + add | **82.03**, $\sigma = 0.0028$ | **51.20**, $\sigma = 0.0112$ | **87.47**, $\sigma = 0.0041$ | 99.46, $\sigma = 0.0044$ | 98.11, $\sigma = 0.0224$ |
| TF + mul | 61.94, $\sigma = 0.0138$ | 23.29, $\sigma = 0.0967$ | 69.57, $\sigma = 0.0439$ | 97.80, $\sigma = 0.0166$ | 96.07, $\sigma = 0.0349$ |
| TF + concat | 80.26, $\sigma = 0.0029$ | 46.57, $\sigma = 0.0108$ | 78.49, $\sigma = 0.0078$ | 99.01, $\sigma = 0.0068$ | 99.46, $\sigma = 0.0051$ |
| BoW + add | **81.90**, $\sigma = 0.0041$ | 40.17, $\sigma = 0.0056$ | 77.62, $\sigma = 0.0176$ | 99.24, $\sigma = 0.0061$ | 96.88, $\sigma = 0.0220$ |
| BoW + mul | **82.59**, $\sigma = 0.0016$ | 50.39, $\sigma = 0.0270$ | 82.22, $\sigma = 0.0168$ | 98.97, $\sigma = 0.0030$ | 32.09, $\sigma = 0.2865$ |
| BoW + concat | 81.57, $\sigma = 0.0013$ | 39.27, $\sigma = 0.0147$ | 79.55, $\sigma = 0.0117$ | 98.79, $\sigma = 0.0056$ | 99.05, $\sigma = 0.0069$ |

*Comparison of HAN an HWAN*

To evaluate our approach, we performed experiments aiming to compare the HAN method with different variants of the HWAN method. It was observed that limiting the length of text processed by the model degraded the performance of the HWAN model to a lesser extent. We also noted that the Reuters dataset, for almost all configurations of the HWAN model, performed better for both restricted and not-restricted document length. The worst results came from the operation of multiplying statistical features and neural network excitation, as can be seen in Tables 3 and 4.

For the limited length of the text processed, the best results were obtained by adding statistical features whose results did not vary much. For the unlimited length of processed text, the best results were achieved for concatenation and addition TFIDF features, with a maximum difference of 0.71 percentage points in their results, and for the addition of TF features.

As shown in Tables 3 and 4, the multiplication operation degraded the resulting representations, which affected the quality of further processing, resulting in a drastic decrease in the performance of these representations in the classification task, especially for the BBCSport dataset.

For further comparison, the TFIDF + *add* variant was denoted as HWAN, as it produced the best performance for the majority of datasets.

For the Reuters dataset, the best performance was achieved by the baseline model BoW (83.42%). The other models performed worse: TFIDF ($-7.84$ percentage point.), LSA ($-19.30$ percentage point.), LDA ($-30.51$ percentage point.), P-SIF ($-20.81$ percentage point.), Doc2Vec-DM ($-3.61$ percentage point.), Doc2Vec-DBoW ($-0.77$ percentage point.), HAN ($-2.09$ percentage point.), HWAN ($-0.57$ percentage point.), Longformer ($-3.52$ percentage point.).

For the Ohsumed dataset, the best performance was achieved by the BoW (60.25%) model. The other models performed worse: TFIDF ($-3.81$ percentage point.), LSA ($-24.53$ percentage point.), LDA ($-40.18$ percentage point.), P-SIF ($-25.92$ percentage point.), Doc2Vec-DM ($-1.59$ percentage point.), Doc2Vec-DBoW ($-0.77$ percentage point.), HAN ($-5.86$ percentage point.), HWAN ($-4.61$ percentage point.), Longformer ($-6.37$ percentage point.).

For the 20Newsgroups dataset, the best performance was achieved by the HAN (89.57%) model. TFIDF ($-4.02$ percentage point.) performed slightly worse. The other models performed worse: BoW ($-17.85$ percentage point.), LSA ($-17.81$ percentage point.), LDA ($-28.67$ percentage point.), P-SIF ($-26.16$ percentage point.), Doc2Vec-DM ($-14.32$ percentage point.), Doc2Vec-DBoW ($-7.05$ percentage point.), HWAN ($-0.89$ percentage point.), Longformer ($-8.24$ percentage point.).

For the BBCSport dataset, the best performance was achieved by the HAN and HWAN models (100.00%). A result better than baseline was also achieved by Longformer (−0.68 percentage point.).

For the BBC dataset, the best performance was achieved by the HWAN model (99.69%). A result better than baseline was also achieved by HAN (−0.45 percentage point.) and Longformer (−0.68 percentage point.).

## 8. Discussion

In summary, for three of the five datasets, the HAN and HWAN models performed the best, for the other two the BoW method produced better results. The HAN and HWAN models dominate the other methods in terms of performance due to their attention mechanism. The models use this mechanism to select the most semantically relevant parts of the text and determine the document class based on them, which is similar to what humans do when reading the text. Longformer's results are worse than expected. Intuition suggests that the attention mechanism should improve the quality of text representations and ease classification. Probably due to the longer input sequence, Longformer performs worse than HAN, as shown in Table 1. On the other hand, for the Reuters and Ohsumed collections, simple statistical methods, such as a BoW or TFIDF, perform better. For data with domain-specific vocabulary, the BoW method performs the best, probably due to the absence of some words in the vocabulary of methods based on pre-trained word representations. The LDA model performs significantly worse on bigger datasets with a larger number of classes. Methods grouped under the name doc2vec achieved the best results for the Ohsumed dataset, excluding the baseline. The doc2vec-DBoW model, in particular, performs very well for all datasets, slightly worse than baseline.

Based on the difference in the results for the HAN and HWAN models in Tables 1 and 5, it can be observed that the deterioration of the results for the Ohsumed and BBC datasets is a result of the fact that the relevant information for the classification problem is included in the initial parts of the documents. Increasing the maximum input processed string results in a decrease of accuracy, which involves processing a longer text, making it more difficult for the model to extract significant information.

**Table 5.** Averaged overall classification accuracy (in percent); cross-validation for Ohsumed, BBC and BBCSport, test-train split for Reuters and 20Newsgroups. The maximum length of input data for HAN and HWAN was set at 30 utterances 300 words each. Numbers in bold denotes results better than BoW and TFIDF or the best result.

| Model | Reuters | Ohsumed | 20Newsgroups | BBC | BBCSport |
|---|---|---|---|---|---|
| BoW | **83.42**, $\sigma = 0.0$ | **60.25**, $\sigma = 0.0$ | 71.72, $\sigma = 0.0042$ | 97.03, $\sigma = 0.0077$ | 96.88, $\sigma = 0.0127$ |
| TFIDF | 75.58, $\sigma = 0.0$ | 56.44, $\sigma = 0.0$ | **85.55**, $\sigma = 0.0008$ | **97.84**, $\sigma = 0.0037$ | **98.51**, $\sigma = 0.0117$ |
| LSA | 64.12, $\sigma = 0.0017$ | 35.72, $\sigma = 0.0012$ | 71.76, $\sigma = 0.0053$ | 96.90, $\sigma = 0.0030$ | 97.83, $\sigma = 0.0117$ |
| LDA | 52.91, $\sigma = 0.0093$ | 20.07, $\sigma = 0.0036$ | 60.90, $\sigma = 0.0102$ | 93.08, $\sigma = 0.0056$ | 92.80, $\sigma = 0.0419$ |
| P-SIF | 62.61, $\sigma = 0.0016$ | 34.33, $\sigma = 0.0015$ | 63.41, $\sigma = 0.0057$ | 95.33, $\sigma = 0.0063$ | 94.70, $\sigma = 0.0267$ |
| Doc2Vec-DM | 79.81, $\sigma = 0.0013$ | 58.66, $\sigma = 0.0021$ | 75.25, $\sigma = 0.0075$ | 96.63, $\sigma = 0.0071$ | 97.29, $\sigma = 0.0096$ |
| Doc2Vec-DBoW | 82.65, $\sigma = 0.0006$ | 59.48, $\sigma = 0.0006$ | 82.52, $\sigma = 0.0040$ | 97.35, $\sigma = 0.0046$ | 97.96, $\sigma = 0.0096$ |
| HAN | 81.33, $\sigma = 0.004$ | 54.39, $\sigma = 0.0029$ | **89.57**, $\sigma = 0.0066$ | **99.24**, $\sigma = 0.0046$ | **100**, $\sigma = 0.0$ |
| HWAN | 82.85, $\sigma = 0.0022$ | 55.64, $\sigma = 0.0027$ | **88.68**, $\sigma = 0.0030$ | **99.69**, $\sigma = 0.0011$ | **100**, $\sigma = 0.0$ |
| Longformer base | 79.90, $\sigma = 0.0042$ | 53.88, $\sigma = 0.0077$ | 81.33, $\sigma = 0.0054$ | **98.20**, $\sigma = 0.0028$ | **99.32**, $\sigma = 0.0061$ |
| SOTA | 97.44 [37] | 75.86 [47] | 88.6 [48] | | 99.59 [37] |

For the limited length of the processed text string, the HWAN model performed better than HAN for the Reuters, Ohsumed and BBC datasets, as shown in Table 1, due to the inclusion of TFIDF weights in the word representation, which allowed the relevance of individual words to be highlighted to a greater extent. Potentially, the use of statistical features computed from a large external corpus can improve HWAN performance, as shown by the difference in results in Tables 3 and 4 for unlimited and limited processed text lengths for the HAN and HWAN models, where the statistical model had access to full documents.

Moreover, with this model, it will be possible to obtain good results while keeping the size of the neural network small.

Table 2 shows the trend in model training time. The BoW and TFIDF models, which rely on word counts in the training dataset, take the shortest amount of time to train. Statistical models, such as LSA and LDA, take much longer due to the fact that they perform singular value decomposition and Gibbs Sampling on matrices of features. As the complexity and size of the model increases, so does the training time, which is why the Longformer model takes the longest time to train of the chosen methods, and requires the most resources.

Comparing the training times of the TFIDF model and Longformer, we observe an increase by about 25,000–80,000 times. It is more efficient to train the HAN model, which needs about 2000 times more time to train, and HWAN—about 6000 times more. The doc2vec models require around 100–200 times more time to train compared to TFIDF.

The summary of our comparative study of representations is presented in Table 6, where we indicate the pros and cons of each method.

**Table 6.** Pros and cons of presented methods. Score is calculated as the sum of the accuracy for all datasets.

| Algorithm | Pros | Cons | Score | Rank |
|---|---|---|---|---|
| BoW | works with noisy data | lack of semantic features<br>sparse representation | 409.30 | 7 |
| TFIDF | words with noisy data<br>takes into account frequent words | lack of semantic features<br>sparse representation | 413.92 | 4 |
| LSA | works for a small number of classes<br>dense representation | reducing dimension results in a<br>loss of information | 366.33 | 8 |
| LDA | works for a small number of classes<br>dense representation | rare words disrupt representations | 319.76 | 10 |
| P-SIF | takes into account topic of the words<br>dense representation | out-of-vocabulary words<br>averaging degenerates representations<br>limited text length<br>computationally intensive | 350.38 | 9 |
| Doc2Vec-DM | mapping context of words<br>dense representation | out-of-vocabulary words<br>computationally intensive | 407.64 | 6 |
| Doc2Vec-DBoW | mapping context of words<br>dense representation | out-of-vocabulary words<br>computationally intensive | 419.96 | 3 |
| HAN | selecting relevant words in context<br>dense representation | out-of-vocabulary words<br>limited text length<br>computationally intensive | 424.53 | 2 |
| Longformer | selecting relevant words in context<br>dense representation | limited text length<br>computationally intensive | 412.63 | 5 |
| HWAN | selecting relevant words in context<br>dense representation<br>additional statistic features | out-of-vocabulary words<br>limited text length<br>computationally intensive | **426.86** | 1 |

## 9. Conclusions

In our study, we present the results of the experimental comparison of selected statistical models and neural networks for the document classification task. Both the quality of text representations, measured as classification accuracy, and the average time required to train the model and its inference, were compared. From the experiments, it can be seen that simple statistical models generate quite good numerical representations of the text and require much less time to train and infer the model. Models based on deep neural networks, such as HAN, HWAN and Longformer, generate representations of good quality,

sometimes better than statistical approaches, but require significantly more computational resources and longer training and processing time.

The proposed method (HWAN), based on improving the hierarchical attention networks, achieved the same or better results compared to the baseline statistical methods for four out of the five training datasets. The study shows that the usage of statistical features calculated based on a larger text corpus allows us to improve the quality of generated text representations without increasing the network size.

The attention-based models presented in this article improve the performance of the baseline methods on three out of the five training sets, even reaching 100% accuracy, as the results for the BBCSport dataset show. These models allow an explanation about which passages in the texts prompted a certain classification decision, by reading which words received a high attention value. However, due to the large computational resource requirements and long training time, they have a low ratio of representation quality to training time. Despite the enormous possibilities offered by language models based on layers of attention and transformer architectures in particular, for some tasks, such as document classification, statistical models are still a good choice.

**Author Contributions:** Data curation, Software, Writing—original draft, Project Administration, Validation, Writing—review and editing, Investigation A.W.; Conceptualization, Methodology, Supervision, Writing—original draft, Writing—review and editing, Validation, J.S. All authors have read and agreed to the published version of the manuscript.

**Funding:** The work has been supported partially by funds of the Faculty of Electronic Telecommunications and Informatics of Gdańsk University of Technology.

**Institutional Review Board Statement:** Not applicable.

**Informed Consent Statement:** Not applicable.

**Data Availability Statement:** The data used in the experiments are publicly available. Details have been given in Section 5.1.

**Conflicts of Interest:** The authors declare no conflicts of interest.

## Appendix A. Singular Value Decomposition

The projection function has to be linear and is based on singular value decomposition. Let us denote real matrix $A$ presented in Formula (A1), where $U$ is an orthogonal matrix of size $M \times M$, $V$ is an orthogonal matrix of size $N \times N$, $V^T$ is a transposition of matrix V and $\Sigma$ is a diagonal matrix of size $M \times N$ containing singular values.

$$A = U\Sigma V^T. \tag{A1}$$

Columns of matrix $U$ and $V$ are called, accordingly, left and right singular vectors of matrix $A$. Elements $\sigma$ of matrix $\Sigma$ are sorted in non-increasing order.

The calculation of singular value decomposition consists of two steps: computation of $AA^T$ and $A^TA$. Firstly, matrix $A_\lambda$ is computed on the basis of Formula (A2), where $\lambda$ is unknown and $I$ is an identity matrix.

$$A_\lambda = AA^T - \lambda I. \tag{A2}$$

Afterwards, the determinant of a matrix $A_\lambda$ is calculated, for example, using Gaussian elimination. The obtained determinant is a characteristic equation of the matrix from which roots, denoted by $\lambda_1, \lambda_2, \ldots$, are calculated. Singular vectors are computed by solving Equation (A3), where $X$ is the expected vector.

$$A_\lambda X = 0. \tag{A3}$$

Singular vectors of matrix $A^TA$ are columns of matrix $V$ and singular vectors of matrix $AA^T$ are columns of matrix $U$. Singular values of $\Sigma$ have squared roots of singular values of $AA^T$ and $A^TA$.

## Appendix B. Gibbs Sampling

This algorithm is described by Formula (A4), where $C_{wj}^{MT}$ denotes the frequency of occurrence of word $w$ as $j$-th topic, $\beta$ is the parameter of probability distribution over words, $C_{d_{ij}}^{DT}$ denotes the frequency of occurrence of document $d$ as $j$-th topic and $\alpha$ is a parameter of probability distribution over the documents.

$$P(z_i = j | z_{-i}, w_i, d_i) \propto \frac{C_{w_{ij}}^{MT} + \beta}{\sum_{w=1}^{M} C_{wj}^{MT} + M\beta} \times \frac{C_{d_{ij}}^{DT} + \alpha}{\sum_{t=1}^{T} C_{d_i t}^{DT} + M\alpha}. \tag{A4}$$

Each iteration of the algorithm updates the probability distributions and frequencies of assignments of words to topics. Parameters $\theta$ and $\phi$, denoting the probability distribution of topic–document and topic–word, are described by Formulas (A5) and (A6), respectively.

$$\phi_i^j = \frac{C_{w_{ij}}^{MT} + \beta}{\sum_{w=1}^{M} C_{wj}^{MT} + M\beta} \tag{A5}$$

$$\theta_i^d = \frac{C_{dj}^{DT} + \alpha}{\sum_{t=1}^{T} C_{dt}^{DT} + T\alpha}. \tag{A6}$$

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
