# Peer review of "Study of Statistical Text Representation Methods for Performance Improvement of a Hierarchical Attention Network"

_applsci, doi:10.3390/app11136113_

Round 1

Reviewer 1 Report

General remarks:

  • This paper is difficult to read. Authors should improve the presentation/description of their proposal.
  • It is not clear the main objective and the scope of this paper:
    • Abstract: “In our work, we study the quality of the text representations using 5 statistical methods and compare them to approaches based on neural networks”
    • Introduction: “The motivation for the study was to find out to what extent the use of deep neural networks improves the quality of representations of texts and how the improvement of the quality influences computation time.”

Neither the abstract nor the introduction refer to interpretability, however, Section 5.2 is dedicated to describing the Local Interpretable Model-agnostic Explanations (LIME) algorithm and does not establish a link to the purpose of the paper. What is the objective of presenting this measure, if it is not used in the evaluation? In Section 6, different representation approaches are compared taking into account accuracy classification and execution times.

  • Paper outline:
    • The outline of the paper is not clear enough
    • There are unnecessary division into subsections, e.g., 1.1-5.1.5
  • Extensive editing of the English language and style is required. Some errors were highlighted in yellow in the revised manuscript. For example:
    • The selected statistical models include Bag of Words (BoW) and Term Frequency - Inverse Document Frequency (TFIDF) weighting, Latent Semantic Analysis (LSA), Latent Dirichlet Allocation (LDA). -> The selected statistical models include Bag of Words (BoW) and Term Frequency - Inverse Document Frequency (TFIDF) weighting, Latent Semantic Analysis (LSA), and Latent Dirichlet Allocation (LDA).
    • In other studies, evaluation is performed for texts oriented on the particular domain. -> In other studies, evaluation is performed for texts oriented on a particular domain.
    • LSA[11] is statistical method used for high dimensional data. -> LSA [11] is a statistical method used for high dimensional data.
    • It performs singular value decomposition of the original matrix, in the case of documents it is a word-document co-occurrence matrix, which results in a more dense, lower-dimensional, representation, called latent semantic space. -> It performs singular value decomposition of the original matrix, in the case of documents it is a word-document co-occurrence matrix, which results in a more dense, lower-dimensional representation, called latent semantic space.
  • Unclear notation: D = d1, d2, . . . , dN, W = w1,w2, . . . ,wM, Z = z(di,wj)i,j
  • Numbers below 10 should be writing with words
  • Presentation and reference to equations are informal and heterogeneous.
    • Wrong style: “Formula 2. where”, “Equation 3. where”, “formula 4. where”, “vector. where”, “Formula 20. Where”, and “representations. where”.
    • Equations 9-12, 14-19 are not referenced
  • Design, presentation, and reference of figures should be improved
    • Figures 1-2, 8-17 are not referenced in the text
    • It is not possible to read numbers and texts in Figures 8-17
  • Tables and figures that describe the main characteristics, advantages, and disadvantages of the representation methods should be included.
  • Unclear statements and terms:
    • “It can be seen from Table 1 that the statistical methods achieve comparable results to the selected neural methods.” How to arrive at that conclusion without performing a statistical analysis? What classifier was used? Furthermore, the results are far from those reported in the SOTA
    • Partition - Smooth Inverse Frequency or Partition Smooth Inverse Frequency?
    • “Dictionary of size V” or “Dictionary V”?
  • Some acronyms are not defined, e.g., P-SIF, BERT, and LSTM
  • Some references are missing, e.g., Singular value decomposition, Gibbs sampling algorithm, Partition Smooth Inverse Frequency, Distributed Memory Model of Paragraph Vectors, Paragraph Vector Distributed Bag of Words method, BERT model, Scaled Dot-Product Attention function.

Author Response

1."This paper is difficult to read. Authors should improve the presentation/description of their proposal."

Presentation and description was clarified.

2. "It is not clear the main objective and the scope of this paper:

Abstract: “In our work, we study the quality of the text representations using 5 statistical methods and compare them to approaches based on neural networks”

Introduction: “The motivation for the study was to find out to what extent the use of deep neural networks improves the quality of representations of texts and how the improvement of the quality influences computation time.”

The conclusion and results of the HWAN model were added to the abstract as suggested. The article's contribution was added in the introduction as suggested.

3. "Neither the abstract nor the introduction refer to interpretability, however, Section 5.2 is dedicated to describing the Local Interpretable Model-agnostic Explanations (LIME) algorithm and does not establish a link to the purpose of the paper. What is the objective of presenting this measure, if it is not used in the evaluation? In Section 6, different representation approaches are compared taking into account accuracy classification and execution times."

LIME algorithm was removed from paper as suggested.

4. "The outline of the paper is not clear enough"

Same as for point 2.

5. "There are unnecessary division into subsections, e.g., 1.1-5.1.5"

Division into subsections makes text more readable and improves information flow.

6. "Extensive editing of the English language and style is required. Some errors were highlighted in yellow in the revised manuscript. (...)"

Erros, typos and style improvements were applied.

7. "Unclear notation: D = d1, d2, . . . , dN, W = w1,w2, . . . ,wM, Z = z(di,wj)i,j"

Notation was clarified. Every variable is explained in place of introduction in text.

8. "Numbers below 10 should be writing with words"

Changed numbers into words as suggested.

9. "Presentation and reference to equations are informal and heterogeneous.

Wrong style: “Formula 2. where”, “Equation 3. where”, “formula 4. where”, “vector. where”, “Formula 20. Where”, and “representations. where”."

Presentation and references of equations were fixed.

10. "Equations 9-12, 14-19 are not referenced"
Equations references were added as suggested.

11. "Design, presentation, and reference of figures should be improved

Figures 1-2, 8-17 are not referenced in the text"

References of figures were added and fixed as suggested.

12. "It is not possible to read numbers and texts in Figures 8-17"
Those numbers are relatively small but can be read when page is in full screen.

13. "Tables and figures that describe the main characteristics, advantages, and disadvantages of the representation methods should be included."

This table already exists. It is Table 6.

14. "Unclear statements and terms: 
“It can be seen from Table 1 that the statistical methods achieve comparable results to the selected neural methods.” How to arrive at that conclusion without performing a statistical analysis? What classifier was used? Furthermore, the results are far from those reported in the SOTA"

Classifier used was multiclass SVM. It was mentioned in the beginning of section 5.

15. "Partition - Smooth Inverse Frequency or Partition Smooth Inverse Frequency?"

Clarified: it is "Partition - Smooth Inverse Frequency".

16. "“Dictionary of size V” or “Dictionary V”?"

Clarified: it is "Dictionary V of size M".

17. "Some acronyms are not defined, e.g., P-SIF, BERT, and LSTM"

Acronyms definitions were added as suggested.

18. "Some references are missing, e.g., Singular value decomposition, Gibbs sampling algorithm, Partition Smooth Inverse Frequency, Distributed Memory Model of Paragraph Vectors, Paragraph Vector Distributed Bag of Words method, BERT model, Scaled Dot-Product Attention function."

Those concepts are referenced in text.

Reviewer 2 Report

Review of Information Manuscript ID: 1216547
Study of Statistical and Neural Text Representation Methods for Performance Improvement of Hierarchical Attention Network

This reviewer finds the paper to be well-written and an important study of NLP techniques for analyzing documents. The conclusions have been well qualified with analysis. The paper has been well-written and flows well. This reviewer appreciates the thorough analysis of the different machine learning methods (both shallow and deep) for NLP including analysis of computing time. There are a few minor typos and grammatical errors, and this reviewer recommends that the authors revise the manuscript to remove them. This reviewer is happy to recommend acceptance of this paper once the grammatical and typographic errors are corrected for publication to Information journal.

Author Response

Grammatical errors and typos have been corrected.

Reviewer 3 Report

The paper ‘Study of Statistical and Neural Text Representation Methods for Performance Improvement of Hierarchical Attention Network“ studies different text representation methods for the document classification task. Authors propose a novel Hierarchical Weighted Attention network (HWAN) that outperforms other studied methods. The paper presents relevant research results that could be of the interest to NLP community. Still, there are several possibilities to improve the reported research.

General remarks:

  1. The Paper is very long. I suggest you consider leaving the only necessary text in sections 2.3 and 2.4 and remove equations to the Appendix (Supplementary material). Equations omitted from section 2.2. can be included in Appendix as well.
  2. Please, include the main finding on HWAN in the abstract with reported results.
  3. Please, include the paragraph on the main scientific contributions in the Introduction. HWAN is the strong point of the paper.
  4. In section 2.2. it is not clear if BOW is constructed over the Boolean model and/or what is the difference between BOW and TFIDF model. This issue will probably be solved by extending the section in Appendix as suggested above. 
  5. I recommend reconsidering the used terminology of utterances: usually, utterances are associated with spoken segments (naturally pronounced sounds in one block within physiological pauses). For text, it is not common to segment into utterances, so please consider changing into sentences that are common in written text or to text segments if you are not dealing with tokenization at the sentence level. 
  6. Section 5: It is not evident why you decide to limit the quantity of the input text to the first 30 sentences and 100 words? Why do you use less data than available? This is not clear even after reading the discussion and explanation of these experiments. So I suggest to elaborate this decision and obtained results better. The same goes for the decision on using the 99th percentile. 
  7. Figures 8, 9, 10, 11, 12, 13, 14 and 15 can be moved to Appendix. Instead, include one table with reported all datasets statistics. 
  8. Section 5.2. I cannot link reported results to LIME evaluation. Only Accuracy is reported in Tables 1, 3, 4. I suggest to either expand with LIME results and discussion or omit LIME from the manuscript. 
  9. Section 6: results - Was accuracy calculated as per class accuracy? Please clarify in the manuscript. 
  10. The second issue to clarify is: why only accuracy was used for evaluation and not AUROC or F1 score. I refer the authors to check discussion regarding using different evaluation metrics for text classification tasks simultaneously to https://www.mdpi.com/2076-3417/9/4/743. 
  11. Next, used datasets significantly differ in the number of classes (91, 23, 20, 5, 5), influencing the obtained results. I suggest including a discussion on this issue as well. It is indicative that for the Reuters with 91 classes, baseline TFIDF /BOW achieves the best-reported results. 
  12. In section 7.1. Listed results in the last paragraph should be reported in a table. What does p.p. mean? 
  13. Also, it is not clear what is the difference between TFIDF+add, +mul and +concat variants? So please include the description. 
  14. In section 8 in Table 6: I suggest adding an additional column to report achieved Rank. 

Specific remarks:

  1. Please correct punctuations (: , . ) in all sentences with equations: (12, 15, 16, 17, 18, …).
  2. Links to all used sw, libraries, github and resources can be moved to references (longformer, papers with code, …) 

Author Response

1. Equations from section 2 were removed to Appendix as suggested.
2. The conclusion and results of the HWAN model were added to the abstract as suggested.
3. The article's contribution was added in the introduction as suggested.
4. Difference between BoW and TFIDF was explained - for BoW, the value is the number of occurrences of the word in the corpus.
5. Word "utterance" was replaced with "sentence" as suggested.
6. In section 5 explanation was added. Input length was limited due to fixed size of neural model architecture and hardware constraints.
7. I think figures that describe datasets are more helpful to the reader. It helps notice clusters in the histogram and to observe outliers, which would be more difficult to spot with the table of statistics.
8. According to suggestion that article is too long description of LIME algorithm was removed.
9. Reported metric was clarified. It is average overall accuracy - for each iteration of training and evaluation percentage of correct predictions was accumulated and averaged.
10. All selected SOTA models were reported with accuracy metric. To fairly compare the models the same metric was used.
11. I think it would be relevant to describe this observation. However, due to the length of the article, it was not included.
12. "p.p." denotes percentage point. Abbreviation was replaced with full name.
13. The model naming convention explanation was added in Section 7.
14. Column "Rank" was added to Table 6.

1. Punctuation from equations was removed.
2. Links were removed to references.